# Laser-Assisted Growth of Carbon-Based Materials by Chemical Vapor Deposition

Abiodun Odusanya [1], Imteaz Rahaman [2], Pallab Kumar Sarkar [2], Abdelrahman Zkria [3], Kartik Ghosh [1] and Ariful Haque [2,4,*]

1   Department of Physics, Astronomy, and Materials Science, Missouri State University, Springfield, MO 65897, USA; abiodun234@live.missouristate.edu (A.O.); kartikghosh@missouristate.edu (K.G.)
2   Electrical Engineering, Ingram School of Engineering, Texas State University, San Marcos, TX 78666, USA; imteaz.rahaman@txstate.edu (I.R.); vlw57@txstate.edu (P.K.S.)
3   Department of Applied Science for Electronics and Materials, Kyushu University, Fukuoka 816-8580, Japan; abdelrahman_zkria@kyudai.jp
4   Materials Science, Engineering, and Commercialization Program, Texas State University, San Marcos, TX 78666, USA
*   Correspondence: ahaque@txstate.edu

**Abstract:** Carbon-based materials (CBMs) such as graphene, carbon nanotubes (CNT), highly ordered pyrolytic graphite (HOPG), and pyrolytic carbon (PyC) have received a great deal of attention in recent years due to their unique electronic, optical, thermal, and mechanical properties. CBMs have been grown using a variety of processes, including mechanical exfoliation, pulsed laser deposition (PLD), and chemical vapor deposition (CVD). Mechanical exfoliation creates materials that are irregularly formed and tiny in size. On the other hand, the practicality of the PLD approach for large-area high-quality CMB deposition is quite difficult. Thus, CVD is considered as the most effective method for growing CBMs. In this paper, a novel pulsed laser-assisted chemical vapor deposition (LCVD) technique was explored to determine ways to reduce the energy requirements to produce high quality CBMs. Different growth parameters, such as gas flow rate, temperature, laser energy, and deposition time were considered and studied thoroughly to analyze the growth pattern. CBMs are grown on Si and Cu substrates, where we find better quality CBM films on Cu as it aids the surface solubility of carbon. Raman spectroscopy confirms the presence of high-quality PyC which is grown at a temperature of 750 °C, $CH_4$ gas flow rate of 20 sccm, a laser frequency of 10 Hz, and an energy density of 0.116 J/cm$^2$ per pulse. It is found that the local pulsed-laser bombardment helps in breaking the carbon-hydrogen bonds of $CH_4$ at a much lower substrate temperature than its thermal decomposition temperature. There is no significant change in the 2D peak intensity in the Raman spectrum with the further increase in temperature which is the indicator of the number of the graphene layer. The intertwined graphene flakes of the PyC are observed due to the surface roughness, which is responsible for the quenching in the Raman 2D signal. These results will provide the platform to fabricate a large area single layer of graphene, including the other 2D materials, on different substrates using the LCVD technique.

**Keywords:** CBMs; laser; graphene; 2D materials; LCVD

## 1. Introduction

Over the last few decades, CBMs have been studied thoroughly in order to gain a better understanding of their adsorption, processes, and isotherms [1]. They are the most widely used electrodes due to their low cost, variety of forms (powders, fibers, aerogels, composites, sheets, monoliths, tubes, etc.), ease of processing, relatively inert electrochemistry, and controllable porosity [2,3]. Activated carbon (AC), graphene, carbon nanotubes (CNTs), carbon nanofibers (CNFs), biochar (BC), and carbon aerogels (CAs) are

all examples of carbon-based materials. Graphene nanomaterials (GFNs) are less toxic than other carbon-based nanomaterials. Their highly flexible physicochemical properties allow the molecule's various functional groups to be altered and perform specific functions [4]. Its sp$^2$ hybridized atoms are held together in a hexagonal honeycomb lattice, and each carbon atom is tightly bound to its three neighboring atoms which can be wrapped into a CNT. The authors present a summary of the latest studies on laser-assisted synthesis of graphene-based materials, along with their development and use as electrodes in supercapacitor and battery applications [5].

Graphene can be synthesized by mechanical exfoliation, liquid-phase exfoliation, reduction of graphene oxide, epitaxial growth and chemical vapor deposition techniques [6–10]. Sample preparation, substrate, reactor dimension, reactor temperature, reactor pressure, gas flow rates, sample position, annealing condition, growing condition, and cooling condition are the critical factors for the growth of graphene by CVD technique [11,12]. Conventionally the hydrothermal and heat treatment methods are suitable for nanoparticle synthesis. The hydrothermal method also takes a lot of time, and the process usually consists of multistage processing steps which is not suitable for mass-scale production [13,14]. The LCVD is a unique CVD technique where the global heat source for the furnace is replaced with a localized laser-heated spot. There are mainly two types of LCVD: pyrolytic and photolytic [15]. The pyrolytic approach is very popular nowadays for producing carbon nanomaterials (CNMs). The fabrication of advanced application materials such as CNTs is the most common use of the pyrolysis process. Usually, it involves a two-stage process: firstly, the precursors for nanomaterials are generated by pyrolysis, while in the second stage, these precursors are usually deposited on nickel-iron and cobalt [16]. The LCVD differs from conventional CVD where the growth area can be limited to the area through which the laser beam passes. Low temperature, shorter reaction times, environmental friendliness, energy savings, catalyst-free growth on insulating substrates, high productivity, improved reproducibility, scalability, and excellent control over experimental parameters are the advantages of laser-based techniques compared to the other conventional methods [17]. The CVD technique is commonly used to make pyrolytic carbons. The PyC structure is a graphite-like, poorly crystalline pure elemental carbon structure. Over the last few decades, pyrolytic carbon has opened a wide scope of new applications including electrochemical measurements, conductive vias through dielectric, carbon-silicon Schottky-boundary diodes and safeguarding of electromagnetic radiation in a wide spectral range.

PyCs are made using the CVD process by heating a hydrocarbon at a high temperature range between 1200 and 1400 °C in the absence of oxygen [18]. The fluidized bed chemical vapor deposition (FBCVD) technique is used to produce PyC even at a higher temperature range between 1250 and 1450 °C. As the deposition temperature increases, the density of PyC drops. In these studies, the researchers were unable to reduce the high temperature requirement in the production of PyCs, and they also failed to account for the effect of deposition time [19]. PyC thin films were produced on SiO$_2$ substrates by CVD at 950 °C under 20 Torr pressure for 30 min. Temperature, pressure, and deposition time were tuned to adjust the thickness and roughness, but this research was conducted only on one type of substrate. Moreover, the deposition time is long which does not help with the reduction in thermal budget. The effect of different substrates was not analyzed [20]. In another study, Hu et al. made PyC at a very high temperature of around 1100 °C. However, the impact of varying deposition time and CH$_4$ flow rate is not addressed during the formation of CBMs [21]. To investigate the effect of temperature, CBMs were grown at three different system temperatures, i.e., 927, 1527, and 2127 °C. High temperatures obviously hasten the formation of carbon structures. However, the temperature of deposited material is exceptionally high [22]. Barberio et al. presented a laser-plasma-driven approach for producing carbon-based nanomaterials, specifically bi- and few-layer graphene. After 10 s of laser irradiation, the number of layers gradually increases from one to two, and after 40 s, the material transforms into graphite. However, the effect of deposition temperature is not considered here [23]. The pyrolysis of hydrocarbons is a complex process with a large

number of different reaction pathways. Ali et al. produced PyC on Si by CVD from gaseous hydrocarbon precursors at about 1000 °C which consists of small, lamellar graphitic ribbons with the size of several nanometers and relatively small amounts of amorphous carbon [24]. Another study shows that Ni–Cu alloy compared to Si is more suitable at low temperatures due to its catalytic nature, low carbon solubility, and more uniform grain size [25]. The film obtained by the proposed method exhibited excellent uniformity and a high monolayer ratio [26]. Only a few studies have been undertaken to reduce the deposition temperature of the CVD process for the growth of carbon-based materials such as graphene. Using thermal CVD, graphene is typically grown between 1000 and 1050 °C [27]. Laser CVD (LCVD) has a lot of promise for local graphene manufacturing using carbon-based materials [28]. The use of a pulsed laser as a secondary source of energy could be a viable option for lowering the deposition temperature and thereby cost to grow CBMs using the CVD technique [29].

To address the aforementioned problems of high energy requirements for CBM production, we studied the growth of CBMs at a low thermal budget by constructing the LCVD system that includes a laser source, mass flow controllers (MFCs), a gas source, a resistant heater, and a vacuum system. Si and Cu substrates are considered in this research to determine the effect of the substrate on PyC growth and to obtain better quality CBM films. In a word, the effects of the deposition temperature, $CH_4$ flow rate, and laser frequency on the growth of CBMs were studied in this research paper to determine the optimum growth conditions.

## 2. Experimental Setup

### 2.1. Nd:YAG Laser

A Nd:YAG laser was used for this research work. The Nd:YAG laser system has four energy levels and can be built in both pulsed and continuous modes. A Xenon or Krypton flash tube is employed as a pumping source [30]. The pulse duration of this laser is 5–7 ns. The principal components of this system are: a traditional pulse laser deposition (PLD) chamber, a solid-state Nd:YAG laser of wavelength 266 nm, mass flow controllers (MFC), a roughing and turbomolecular pump, a baratron gauge, a bourdon gauge, a heater, a temperature controller, a UV mirror, a network of gas supply lines, and the precursor gases (mixture of $H_2$ and Ar gas). For this experiment, two systems were designed as the primary sources of heating: the first one is the use of resistive heating in the chamber and the second one is inductive heating using a quartz tube in an induction furnace. To initiate a deposition, the substrates are placed in the deposition chambers, and it is pumped down from atmospheric pressure to about $5 \times 10^{-5}$ Torr. Pulsed laser beam incidents on the copper and silicon substrates occurred with the aid of a 248 nm UV mirror for different deposition times of 1–5 min. Figure 1 depicts the deposition process in the chamber.

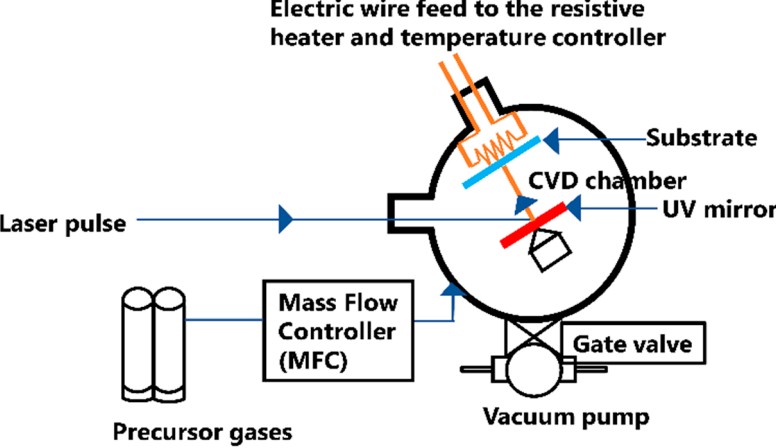

**Figure 1.** Schematic diagram of the deposition process in the chamber.

## 2.2. Characterization: Raman Spectroscopy

In this research, the Raman spectrum was produced using a Horiba Labram Raman-PL system at room temperature. As an excitation source, a green laser with a wavelength of 532 nm was used on the sample.

## 3. Result Analysis

In this study, silicon dioxide wafer ($Si_{x,y}$) and copper foil ($Cu_{x,y}$) were used as substrates where x represents the sample number. Here, y is '0' when there is no laser irradiation and '1' when there is laser irradiation. The deposition pressure was maintained at 550 Torr, $Ar/H_2$ flow rate was set to 10 sccm, and the energy density was calculated to be 0.116 J/cm² per pulse. In this research, the $CH_4$ flow rate, temperature, substrate, and time were mainly varied to observe the overall effects of the process parameter.

### 3.1. Variation in CH₄ Flow Rate and Temperature

The deposition is undertaken with two different flow rates of $CH_4$: 10 and 20 sccm. Here, the deposition temperature is varied between 650 and 850 °C and the effect of temperature change is observed. Before we proceed, the Raman spectra for commercial grade graphene obtained from the Airforce Research Laboratory are shown in Figure 2.

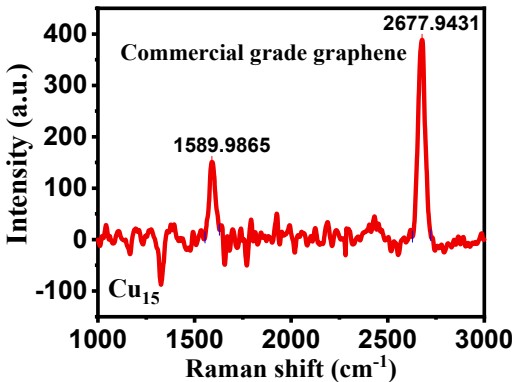

**Figure 2.** Raman spectra of pristine high−quality graphene.

### 3.1.1. Constant CH₄ Flow Rate at 10 sccm and Varied Temperature

At first, we kept a constant $CH_4$ flow rate of 10 sccm and varied the temperature between 650 and 850 °C using copper substrates. The laser frequency was likewise changed between 5 and 10 Hz, with a 5 min deposition duration. Figure 3 shows the Raman spectra considering all these parameters.

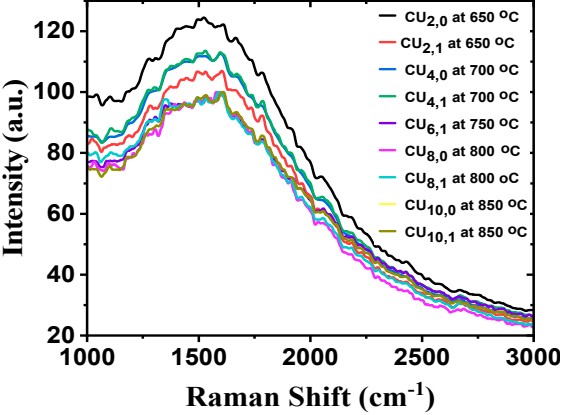

**Figure 3.** Raman spectra at a constant $CH_4$ flow rate at 10 sccm and different deposition temperatures.

It is observed that there are no distinguishable peaks to indicate crystalline carbon growth. According to Grove's model, the mass transfer of reactant gaseous species is solely dependent on mass diffusion. As a result, there is a concentration gradient of gaseous species, and the flux of mass transport from the gas phase is inadequate to activate the necessary reaction at the substrate surface. This indicates that the given flow rate of 10 sccm is insufficient for deposition to occur. Furthermore, the requisite temperature to thermally break the carbon–hydrogen bonds (413 kJ/mol) at the required dehydrogenation energy is not met.

### 3.1.2. Constant $CH_4$ Flow Rate at 20 sccm and at Different Temperatures

Figure 4 shows the Raman spectra graphs for copper and silicon substrates, respectively. At the same time, we have adjusted the temperature while maintaining a constant $CH_4$ flow rate of 20 sccm and a laser frequency of 10 Hz.

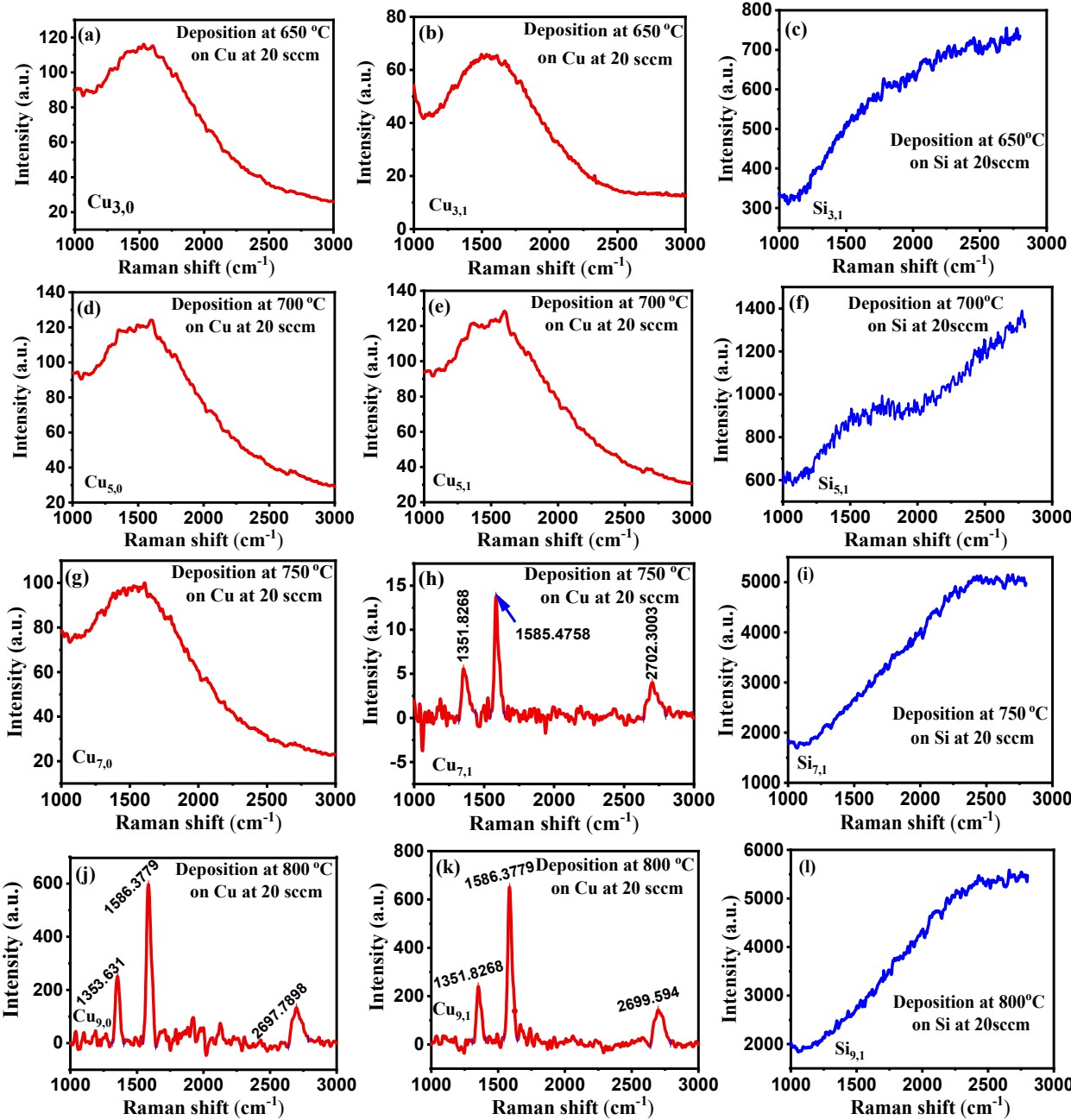

**Figure 4.** *Cont.*

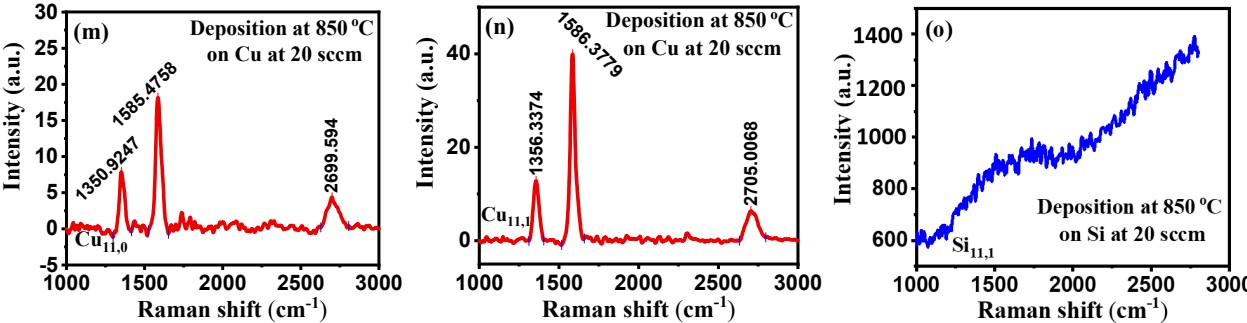

**Figure 4.** (**a–o**) Raman spectra at constant CH$_4$ flow rate at 20 sccm with different deposition temperatures on Cu and Si substrates.

As the CH$_4$ flow rate of 10 sccm is too low for the deposition, we increased the CH$_4$ flow rate to 20 sccm and repeated the experiment. From Figure 4, it is observed that SiO$_2$ substrate has no distinguishable peak to ensure the deposition of any type of CBM in crystalline form. Similarly, there are no identifiable peaks for the copper substrate at 650 °C, but a very slight one is observed on the film grown at 700 °C. However, we found three peaks at 1351.8, 1585.5, and 2702.3 cm$^{-1}$ on the laser incident part of our sample at 750 °C, which are D, G, and 2D peaks, respectively. These are the characteristic peaks for carbon-based materials such as graphene. We further raised the temperature of the resistive heater to 850 °C at an increment rate of 50 °C and found CBMs in both laser incident and non-incident areas. The Raman shifts for the D, G, and 2D peaks were all in the same ballpark. This means that at 20 sccm, we were able to commence the reaction by generating enough mass diffusion and flux from the gas phase and the substrate surface. Furthermore, the pulsed laser photons effectively raised the local temperature of the bombarded region. Under these deposition conditions, our CBMs were highly defective due to the predominance of the D peak intensity compared to pristine graphene.

### 3.2. Variation in Deposition Time

Here, we varied the deposition time and kept other parameters constant as shown in Table 1 and Figure 5a–f.

**Table 1.** Experimental parameters of different deposition times with constant CH$_4$ flow rate at 20 sccm and 850 °C deposition temperature.

| Sample ID | Process Parameters | | | |
| --- | --- | --- | --- | --- |
| | Temp (°C) | Laser Pulse Frequency (Hz) | CH$_4$ Flow Rate (sccm) | Deposition Time (min) |
| Cu$_{11,0}$ | 850 | 10 | 20 | 5 |
| Cu$_{11,1}$ | 850 | 10 | 20 | 5 |
| Cu$_{12,0}$ | 850 | 10 | 20 | 3 |
| Cu$_{12,1}$ | 850 | 10 | 20 | 3 |
| Cu$_{13,0}$ | 850 | 10 | 20 | 1 |
| Cu$_{13,1}$ | 850 | 10 | 20 | 1 |

Our main goal was to shorten the CBM deposition time and sharpen the 2D peak by reducing the number of layers. The deposition time was reduced from 5 to 3 and then to 1 min, but there was no significant improvement in the 2D peak. From this, it is concluded that it might not be graphene but rather a graphene-like material. It was observed that the deposited substrate was PyC. It is produced by heating a hydrocarbon (CH$_4$) nearly to its thermal decomposition temperature of around 1000–1200 °C. This thermal decomposition temperature breaks its bonds, releasing carbon-free radicals and allowing graphite to crystalize in the absence of oxygen (pyrolysis) [31]. The surface roughness of PyC films

does not vary significantly with film thickness [32], and it is also responsible for quenching the 2D signal in samples $Cu_{11,1}$, $Cu_{12,1}$, and $Cu_{13,1}$, regardless of the deposition time.

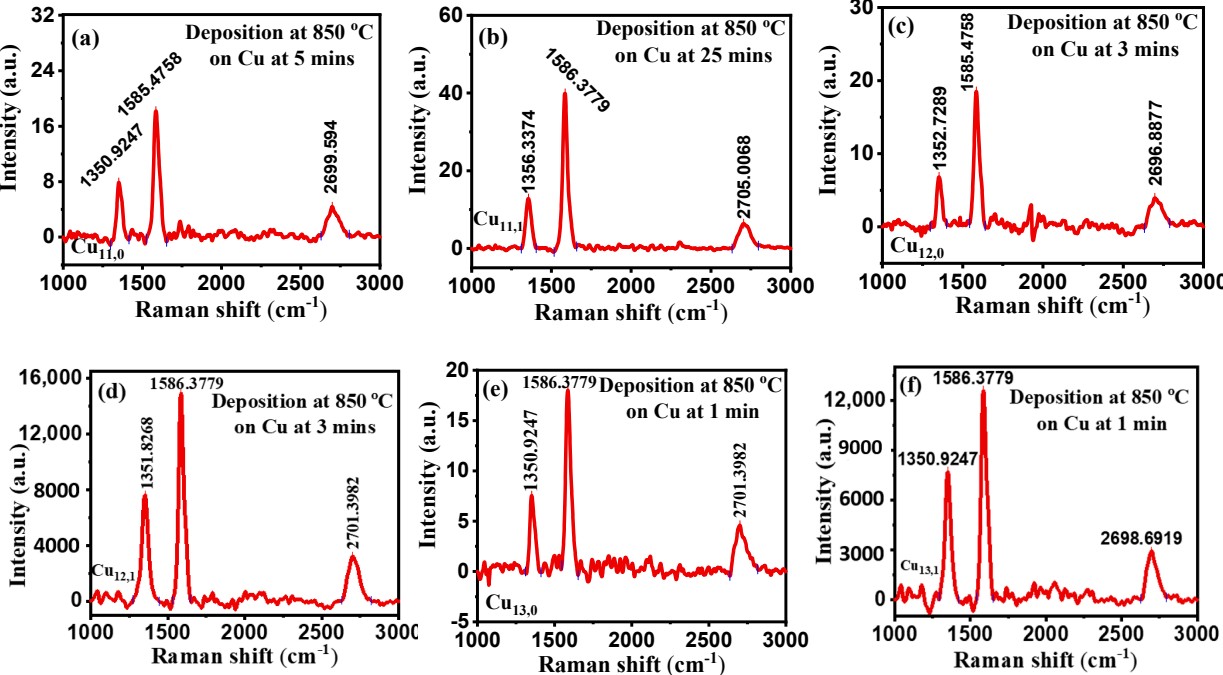

**Figure 5.** (**a**–**f**) Raman spectra at different deposition times with a constant $CH_4$ flow rate of 20 sccm and 850 °C deposition temperature.

It was observed that our CBMs were highly defective because of the prominence of the intensity of the D peak when compared with high quality graphene in Figure 2. Sample $Cu_{15}$ is the commercial-grade graphene that we purchased and used as the gold standard for our deposition. Here, we can see that it is a single layer graphene with zero defects. It can be seen from the D peak that sample $Cu_{13,1}$ (Figure 5f) has the highest defect concentration, the defect data looks rather inconclusive, and more research needs to be undertaken on this; the same goes for the carbon concentration shown from the G peak.

### 3.3. Effects of Laser Energy on PyC Deposition

Here, we compared two depositions, one grown by purely thermal CVD and the other by LCVD. The process parameters are shown in Table 2, and the Raman data plots are shown in Figure 6a–c below.

**Table 2.** Effect of laser energy on PyC deposition.

| Sample ID | | Process Parameters | | | |
|---|---|---|---|---|---|
| | | Temp (°C) | Laser Pulse Frequency (Hz) | $CH_4$ Flow Rate (sccm) | Deposition Time (min) |
| $Si_{11,1}$ | $Cu_{11,0}$ | 850 | 10 | 20 | 5 |
| | $Cu_{11,1}$ | 850 | 10 | 20 | 5 |
| | $Cu_{14,0}$ | 850 | 0 | 20 | 5 |

Using samples $Cu_{11}$ and $Cu_{14}$, we evaluated the effect of a pulsed laser on the formation of PyC. In both the laser and non-laser bombarded regions of sample $Cu_{11}$, we found the D, G, and 2D peaks. This implies that the laser shots generate a local laser bombardment region and act as a complementary energy source, which raises the local temperature of the laser spot. This raise in temperature provides the additional energy required to initiate the

reaction and increases the deposition area. On the other hand, the thermal energy supplied by the resistive heater was insufficient to initiate the PyC deposition on $Cu_{14}$ sample.

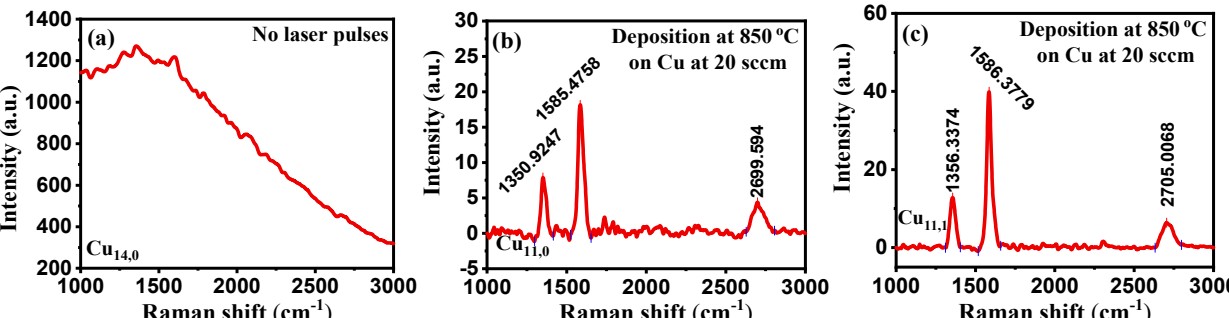

**Figure 6.** (**a–c**) Raman data plots showing the effect of: (**a**) no laser pulses; (**b**) laser incident spot; (**c**) laser non-incident spot on PyC deposition.

### 3.4. Effects of the Different Substrates on PyC Deposition

The performance of silicon and copper substrates in PyC deposition was compared at 850 °C with process parameters set to 10 Hz, $CH_4$ flow rate of 20 sccm, and deposition time of 5 min. The Raman plots are shown, considering all these conditions, in Figure 4 (n, o). Although there is no discernible peak in the Raman spectra for the silicon substrate, which indicates the presence of PyC, the bump-like shape of the graph indicates the presence of amorphous carbon. On the other hand, copper substrate exhibits crystalline $sp^2$ hybridization in the form of PyC which is ensured by the D, G, and 2D peaks in the Raman spectrum. As a result, the copper substrate acts as a catalyst for PyC deposition by aiding the surface solubility of carbon on copper.

### 3.5. Reaction Mechanism for PyC Deposition

Taking a cue from Gajewski, Grzegorz, and Pao, Chun-Wei [33], we investigated the chemical pathways for $CH_4$ decomposition and dehydrogenation on the copper substrate surface. Multiple ground and transition states were used in this study since the dehydrogenation process is a multi-step process with a series of intermediates. This reaction had six ground states (GS) and four transitional stages (TS), as indicated below:

$GS1–CH_4$
$GS2–CH_3 + H$
$GS3–CH_2 + 2H$
$GS4–CH + 3H$
$GS5–C + 4H$
$GS6–1/2C_2 + 4H$
$(TS1): CH_4 \rightarrow CH_3 + H$   $(\Delta H = +0.75 \text{ eV}; E_{act} = +1.57 \text{ eV})$
$(TS2): CH_3 \rightarrow CH_2 + H$   $(\Delta H = +0.83 \text{ eV}; E_{act} = +1.36 \text{ eV})$
$(TS3): CH_2 \rightarrow CH + H$   $(\Delta H = +0.41 \text{ eV}; E_{act} = +0.94 \text{ eV})$
$(TS4): CH \rightarrow C + H$   $(\Delta H = +1.22 \text{ eV}; E_{act} = +1.84 \text{ eV})$

In the gas phase, the activation barriers for $CH_4$ decomposition are substantially larger than on the surface of the copper substrate. All the reactions are endothermic except for the formation of carbon. The entire reaction is described as follows:

$CH_4(g) \rightarrow C\ (s) + 4H$   $(\Delta H = +3.20 \text{ eV})$

### 3.6. Quantitative Analysis of Results

The number of layers can be estimated by dividing the peak intensities of D and G or 2D and G [34]. The number of layers can also be calculated using the ratio of the areas of the individual peaks. The estimated ratio of intensities by the number of graphene layers is shown in Table 3.

**Table 3.** The ratio of peak intensities and number of layers.

| Number of Graphene Layers | Approximate Peak Intensities | |
|---|---|---|
| Single layer | $\frac{I_D}{I_G} = 0$ | $\frac{I_{2D}}{I_G} = 2$ |
| Double layer | $\frac{I_D}{I_G} = 0.05$ | $\frac{I_{2D}}{I_G} = 1$ |
| Few layers | $\frac{I_D}{I_G} = 0.1$ | $\frac{I_{2D}}{I_G} = 0.8$ |
| Multi-layer | $\frac{I_D}{I_G} = 0.18$ | $\frac{I_{2D}}{I_G} = 0.5$ |

Table 4 below shows the peak positions, intensity, and the number of layers obtained from each deposition from Raman analyses.

**Table 4.** Peak positions, intensities, and the number of layers from the Raman signal.

| ID | Peak Position | | | Peak Intensity | | | $\frac{I_{2D}}{I_G}$ | Peak Area | | | $\frac{A_{2D}}{A_G}$ |
|---|---|---|---|---|---|---|---|---|---|---|---|
| | D | G | 2D | D | G | 2D | | D | G | 2D | |
| $Cu_{7,1}$ | 1351 | 1585 | 2702 | 5.5 | 14.3 | 4.2 | 0.3 | 267.1 | 548.4 | 377.4 | 0.7 |
| $Cu_{9,0}$ | 1352 | 1586 | 2697 | 237.3 | 612.9 | 142.3 | 0.2 | 8828.0 | 27,855.0 | 11,766.2 | 0.4 |
| $Cu_{9,1}$ | 1351 | 1586 | 2699 | 234.2 | 621.5 | 155.2 | 0.3 | 9723.4 | 24,434.8 | 13,616.0 | 0.6 |
| $Cu_{11,0}$ | 1350 | 1585 | 2699 | 7.8 | 18.3 | 4.2 | 0.2 | 294.9 | 822.5 | 330.7 | 0.4 |
| $Cu_{11,1}$ | 1354 | 1586 | 2704 | 9.1 | 26.0 | 5.1 | 0.2 | 451.4 | 1157.6 | 401.9 | 0.4 |
| $Cu_{12,0}$ | 1351 | 1585 | 2697 | 6.9 | 18.3 | 3.8 | 0.2 | 263.6 | 752.8 | 350.1 | 0.5 |
| $Cu_{12,1}$ | 1352 | 1586 | 2701 | 6858 | 14,769 | 3103 | 0.2 | 320,096.0 | 644,120.2 | 249,714.2 | 0.4 |
| $Cu_{13,0}$ | 1350 | 1586 | 2701 | 8.6 | 18.1 | 4.5 | 0.3 | 427.3 | 792.3 | 365.5 | 0.5 |
| $Cu_{13,1}$ | 1350 | 1586 | 2698 | 7607 | 12,899 | 3008 | 0.2 | 366,476 | 633,137 | 250,835 | 0.4 |
| $Cu_{15}$ | 0 | 1589 | 2677 | 0 | 140.6 | 375.4 | 2.7 | 0 | 5129.9 | 15,718.5 | 3.1 |

We can conclude from Table 4 that all grown samples had many graphene flakes based on the intensity ratios. Instead of looking at individual peaks, the same information can be obtained by looking at the ratio of the areas. The $Cu_{15}$ is a commercial-grade graphene sample that we bought and used as a gold standard for deposition. It is identified that graphene is a single layer with no imperfections. The D peak shows that sample $Cu_{13,1}$ has the highest defect concentration; however, the defect data appears to be equivocal, and more investigation is needed. Additionally, the carbon concentration represented by the G peak is also the same.

## 4. Conclusions

In this paper, the impacts of laser irradiation on the deposition of pyrolytic carbon using optimal deposition parameters on the copper and silicon dioxide substrates were systematically investigated. Copper foil was found to be the more suitable substrate to grow PyC than silicon dioxide wafer for our experimental setup because copper aids the surface solubility of carbon. From Raman data, we concluded that the PyC shares similar characteristics with graphene because of the coinciding Raman shifts of the D, G, and 2D peaks at 1350, 1600, and 2700 $cm^{-1}$, respectively. Furthermore, the PyC is randomly oriented and intertwined with graphene flakes produced by heating a hydrocarbon nearly to its thermal decomposition temperature. Pyrolytic carbon is relatively more defect-ridden than high-quality graphene because of the intertwined nature of its constituent graphene flakes. Despite having graphene as its fundamental building block, the intertwining of the graphene flakes causes some surface roughness that does not vary significantly with film thickness, and it is also responsible for quenching the 2D peak on the Raman spectra regardless of deposition time, as seen in samples $Cu_{11,1}$, $Cu_{12,1}$, and $Cu_{13,1}$.

**Author Contributions:** A.O., I.R. and P.K.S. contributed equally to this work. Data curation A.O., I.R. and P.K.S. Conceptualization, A.O. and I.R. Formal analysis I.R. and P.K.S. Methodology A.Z., Project Administration A.H. and K.G. Resources and supervision A.H. Validation A.Z. Writing— original draft A.O., I.R. and P.K.S. All authors have read and agreed to the published version of the manuscript.

**Funding:** This work was supported by the NSF Grant (DMR-08211593).

**Institutional Review Board Statement:** Not applicable.

**Informed Consent Statement:** Not applicable.

**Data Availability Statement:** Not applicable.

**Acknowledgments:** We would like to acknowledge Rishi Patel, senior research scientist, Jordan Valley Innovation Center (JVIC), Springfield, MO 65806, USA, for his feedback and the useful discussions.

**Conflicts of Interest:** On behalf of all authors, the corresponding author states that there is no conflict of interest.

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
