# Peer review of "Laser-Assisted Growth of Carbon-Based Materials by Chemical Vapor Deposition"

_carbon_

Round 1
Reviewer 1 Report
The manuscript encompasses the technique of pulsed laser CVD where the authors have tried to play with flow rate and deposition temperature. However, the manuscript in its present state lacks some data, therefore the following edits are recommended to the authors to add in the manuscript and resubmit it.
- The SI units should be properly written. It is 650 ℃ not 650℃. Similarly "space" should be exercised before "sccm"
- Space should be appropriately used between two letters. Although comments have been mentioned, authors should take a serious revision of these "space" errors.
- PyC is not a very generalized abbreviation for pyrolytic carbon and hence it is suggested to not be mentioned in the keywords.
- Authors should also present some evidences that the D band intensity is not contributed by the laser i.e. the laser used for Raman studies is not inducing any defects during the measurements. Optical Images before and after the Raman measurement for at least two of Cu and Si wafer must be provided.
- No Raman data is mentioned for the Cu15 sample which the authors have expressed as monolayer Graphene standard.
- The claims like "twining of the graphene flakes causes some surface roughness that does not vary significantly with film thickness" has no evidence to support. Authors must provide relevant AFM and SEM data to verify the hypothesis.
- An XPS study should be performed on both the samples to further ascertain the reasons for the different growth behavior of Pyrolytic carbon on Cu Vs Si. The role of Cu in graphene growth has been outlined very well by several prior studies For eg: https://pubs.acs.org/doi/abs/10.1021/nl201566c?casa_token=PMe34EJiOysAAAAA:UvmU8rsgHS5HeuKQyE0u0Rf-go6tkI2zIB6XxioHHkI1aW_m3I1wc0ykjS8rZ9u4cgo-0pJCWECPMvG1LA
- Discussions about the type of substrate in the form of Cu tapes is missing. The authors must reflect some notations based on the prior literature on the growth of pyrolytic carbon on Cu as compared to Si.
- Authors should also provide a more clearer outlook.

Author Response
Reviewer # 1
1. The SI units should be properly written. It is 650 ℃ not 650℃. Similarly, "space" should be exercised before "sccm"
Response: In the revised manuscript we have provided the SI unit properly, given space before ℃ and sccm.
2. Space should be appropriately used between two letters. Although comments have been mentioned, authors should take a serious revision of these "space" errors.
Response: According to the suggestion, we have revised the error in the revised paper.
3. PyC is not a very generalized abbreviation for pyrolytic carbon and hence it is suggested to not be mentioned in the keywords.
Response: As per the reviewer’s suggestion, we have removed the “PyC” in keyword section on the revised paper.
4. Authors should also present some evidence that the D band intensity is not contributed by the laser i.e., the laser used for Raman studies is not inducing any defects during the measurements. Optical Images before and after the Raman measurement for at least two of Cu and Si wafer must be provided.
Response: Throughout the measurements we have maintained a minimum laser power to attain the spectra. Usually, the radiation induced visible defects are observed during Raman spectroscopy in soft materials. In this study the carbon samples are neither single layer graphene nor carbon-based soft organic material, thus the D band cannot be associated with the laser induced damage in the carbon structure. Moreover, we do not see any visible damage before and after the Raman measurements. And the defects in the samples due to the D band are associated with inhomogeneity/point and planer defects at the atomic scale. Thus, it is impossible to visually identify these defects using optical/scanning electron microscopy.
5. No Raman data is mentioned for the Cu15 sample which the authors have expressed as monolayer Graphene standard.
Response: We have added the Raman plots data of Cu15 as a commercial grade monolayer graphene in Fig. 2.
It was observed that our CBMs were highly defective because of the prominence of the intensity of the D peak when compared with high quality graphene in Figure 2. Sample Cu15 is the commercial grade graphene that we purchased and used as the gold standard for our deposition. Here, we can see that it is a single layer graphene with zero defect. It can be seen from the D peak that sample Cu13,1 (Fig. 5 f) has the highest defect concentration, the defect data looks rather inconclusive, and more research needs to be done on this, same goes for the carbon concentration shown from the G peak.
6. The claims like "twining of the graphene flakes causes some surface roughness that does not vary significantly with film thickness" has no evidence to support. Authors must provide relevant AFM and SEM data to verify the hypothesis.
Response: We plan to perform the suggested experiments and analyses in future, and we will publish the data in the next phase of the research.
7. An XPS study should be performed on both the samples to further ascertain the reasons for the different growth behavior of Pyrolytic carbon on Cu Vs Si. The role of Cu in graphene growth has been outlined very well by several prior studies For e.g.: https://pubs.acs.org/doi/abs/10.1021/nl201566c?casa_token=PMe34EJiOysAAAAA:UvmU8rsgHS5HeuKQyE0u0Rf-go6tkI2zIB6XxioHHkI1aW_m3I1wc0ykjS8rZ9u4cgo-0pJCWECPMvG1LA
Response: We plan to perform the suggested experiments and analyses in future, and we will publish the data in the next phase of the research.
8. Discussions about the type of substrate in the form of Cu tapes is missing. The authors must reflect some notations based on the prior literature on the growth of pyrolytic carbon on Cu as compared to Si.
Response: We have added relevant references on the growth of pyrolytic carbon on Cu and Si in the revised manuscript. The pyrolysis of hydrocarbons is a complex process with a large number of different reaction pathways. The chemistry and kinetics of PyC formation from different hydrocarbons has been extensively investigated by a number of groups. Ali et al. produced PyC by CVD from gaseous hydrocarbon precursors at about 1000 °C which consists of small, lamellar graphitic ribbons with size of several nanometers and relatively small amount of amorphous carbon on black silicon wafer. Another study shows that Ni-Cu alloy compared to Si is more suitable at low temperatures due to its catalytic nature, low carbon solubility and more uniform grain size. The film obtained by the proposed method exhibited excellent uniformity and a high monolayer ratio [24-26].
9. Authors should also provide a clearer outlook.
Response: We have carefully revised the whole paper and the overall quality of the manuscript has been tremendously improved in the revised version.
Reviewer 2 Report
The research article submitted by Odusanya et al. as "Laser assisted growth of carbon-based materials by chemical vapor deposition" in interesting and contains valuable information about growth of carbon based materials by laser-assisted CVD method. After carefully reading, it was found that this article needs major revision because several issues and explanations are still need to be clarified. I recommend it publication in this journal (MAJOR REVISION) after providing proper improvement in revised version by including suggestion, modification and reply to raised queries which are given below.
- What is the main advantage of this synthesized material and how this reported research work is different from other research work? Try to differentiate including synthesis approach, cost-effective and valuable materials.
- How laser-assisted formation of carbon materials is facile and cost effective as compared to other synthesis method as hydrothermal and heat treatment method. Explain in detail in introduction section.
- Compared to liquid hydrocarbon precursor, CH4 gas is expensive, laser unit is expensive then how authors justify that it simple as well as economic to select as synthesis approach?
- Improve the abstract by including the actual finding and novelty of this research work.
- I cannot find a real literature review in introduction section and why authors have done this work, what is the scientific gap, and what is novelty in this submitted research work.
- Using laser based approach, several works has been reported as formation of graphene through laser, and carbonization of polymer using various laser approached. Need inclusion in introduction to shows the updated research as Progress in Energy and Combustion Science (DOI: 10.1016/j.pecs.2021.100981); Coordination Chemistry Reviews 342, 34-79, 2017; RSC Advances 6 (86), 84769–84776, 2016; Journal of Colloid and Interface Science 507, 271-278, 2017; Energy 179, 676-684, 2019 etc.
- How flow rate of CH4 and deposition temperature affect the thickness of carbon materials? Explain in details
- In Raman spectra (Fig. 3(a-o),provide the data from 300 cm-1. There is no appearance od D-band in provided data. How D and G band intensity varied with deposition temperature and CH4 gas flow?
- How 2D band in Raman spectra related with thickness/ No of carbon layers? Provides discussion on this finding.
- Authors should provide the SEM images of deposited carbon materials and analyzed its surface morphology. Only Raman characterization in not sufficient in this study.
- Provided the XRD pattern of carbon materials and check it crystallinity.
- What was the amount of C and O in synthesized carbon materials? Check it by using EDS-SEM analysis.
- There are some grammatical and punctuation errors in this manuscript. The English language should be improved. Tenses are not consistent from sentence to sentence and there are some grammatical errors.
Author Response
- What is the main advantage of this synthesized material and how this reported research work is different from other research work? Try to differentiate including synthesis approach, cost-effective and valuable materials.
Response: In the revised manuscript we have added the following information “Mechanical exfoliation creates materials that are irregularly formed and tiny in size. Impurities are introduced during solution synthesis, lowering performance. The practicality of the laser approach for large-area deposition is still debatable. As a result, CVD is the most effective method for scaled low-cost preparation”. We have provided more insights on this in the introduction as well.
2. How laser-assisted formation of carbon materials is facile and cost effective as compared to other synthesis method as hydrothermal and heat treatment method. Explain in detail in introduction section.
Response: Conventionally the hydrothermal and heat treatment methods are suitable for nanoparticle synthesis. This research is all about thin film fabrication by chemical vapor deposition technique. Hydrothermal and heat treatment methods are not suitable to fabricate for the large area continuous carbon-based thin films. The hydrothermal method also takes a lot of time, and the process usually consists of multistage processing steps which is not suitable for mass scale production [13-14].
- Compared to liquid hydrocarbon precursor, CH4 gas is expensive, laser unit is expensive then how authors justify that it simple as well as economic to select as synthesis approach?
Response: Liquid hydrocarbons are not suitable for clean carbon-based thin film research for electronic applications. Hence the researchers in this field do not consider conventional liquid hydrocarbons such as crude oil, gasoline, natural gas, etc. while realizing the cost/economic aspect of the fabrication. CH4 is a widely used gas to fabricate CBMs such as graphene, reduced graphene oxide, diamond, and so on [1-3]. Although the market price of excimer laser is high (up to $120,000), the average lifetime (around 30 years) of such lasers is very long. Thus, the overall fabrication cost would be reasonably low for mass scale production using the proposed technique.
- Improve the abstract by including the actual finding and novelty of this research work.
Response: As per reviewer’s suggestion, we have modified the abstract by incorporating the following information- ‘CBMs have been grown using a variety of processes, including mechanical exfoliation, pulsed laser deposition, and chemical vapor deposition (CVD). Mechanical exfoliation creates materials that are irregularly formed and tiny in size. On the other hand, the practicality of the pulse laser deposition (PLD) approach for large-area high-quality CMB deposition is quite difficult. Thus, CVD is considered as the most effective method for growing CBMs. In this paper, a novel pulsed laser assisted chemical vapor deposition (LCVD) technique has been explored to find out the ways for reducing the energy requirements to produce CBMs’.
- I cannot find a real literature review in introduction section and why authors have done this work, what is the scientific gap, and what is novelty in this submitted research work.
Response: We have addressed the concern by clarifying the shortcomings of the previous studies. We have modified the introduction section and added relevant information, i.e., “PyCs are made using the CVD process by heating a hydrocarbon at a high temperature range between 1200 to1400 ℃ in the absence of oxygen [18]. Fluidized bed chemical vapor deposition (FBCVD) technique is used to produce PyC even at a higher temperature range between 1250 to 1450 ℃. As the deposition temperature increases, the density of PyC drops. In all these studies the researchers were unable to reduce the high temperature requirement in the production of PyCs, and they also failed to account for the effect of deposition time [19]. PyC thin films were produced on SiO2 substrates by CVD at 950 °C under 20 Torr pressure for 30 minutes. Temperature, pressure, and deposition time were tuned to adjust the thickness and roughness, but this research was conducted only on one type of substrate. Moreover, the deposition time is long which does not help in cost reduction. The effect of different substrates was not analyzed [20]. In another study Hu et al. made Pyrolytic carbon at a very high temperature of around 1100 ℃. However, the impact of varying deposition time and CH4 flow rate is not addressed during the formation of CBMs [21].”
- Using laser-based approach, several works has been reported as formation of graphene through laser, and carbonization of polymer using various laser approached. Need inclusion in introduction to shows the updated research as Progress in Energy and Combustion Science (DOI: 10.1016/j.pecs.2021.100981); Coordination Chemistry Reviews 342, 34-79, 2017; RSC Advances 6 (86), 84769–84776, 2016; Journal of Colloid and Interface Science 507, 271-278, 2017; Energy 179, 676-684, 2019 etc.
Response: We have included the following references. The other publication does not appear to be precisely aligned with our study objectives.
10.1016/j.pecs.2021.100981: The authors present a summary of latest studies on laser-assisted synthesis of graphene-based materials, along with their development and use as electrodes in supercapacitor and battery applications.
Coordination Chemistry Reviews 342, 34-79, 2017: Laser CVD (LCVD) has a lot of promise for local graphene manufacturing using carbon-based materials.
- How flow rate of CH4 and deposition temperature affect the thickness of carbon materials? Explain in detail?
Response:
CH4 flow rate effect including temperature:
First of all, we tried with CH4 flow rate of 10sccm which is insufficient for deposition to occur. In page -5 #3-8 lines clarifies the statements.
“According to Grove's model, the mass transfer of reactant gaseous species is solely dependent on mass diffusion. As a result, there is a concentration gradient of gaseous species, and the flux of mass transport from the gas phase is inadequate to activate the necessary reaction at the substrate surface. This indicates that the given flow rate of 10sccm is insufficient for deposition to occur. Furthermore, the requisite temperature to thermally break the carbon-hydrogen bonds at the required dehydrogenation energy is not met.”
For this reason, we increase the methane flow rate to 20 sccm and repeat the experiment to observe the effects of increasing CH4 flow rate. And we find the presence of high-quality PyC at CH4 gas flow rate of 20sccm, which is grown at a temperature of 750℃, a laser frequency of 10 Hz, and an energy density of 0.116 J/cm2 per pulse.
Temperature Effect: The temperature effect is described in the following segment- P-6; #1-14
“As the CH4 flow rate of 10 sccm is too low for the deposition, we increased the CH4 flow rate to 20 sccm and repeated the experiment. From Fig. 4, it is observed that SiO2 substrate has no distinguishable peak to ensure the deposition of any type of CBM in crystalline form. Similarly, there are no identifiable peaks, for the copper substrate at 650 ℃, but a very slight one at 700 ℃. However, we found three peaks at 1351.8 cm-1, 1585.5 cm-1, and 2702.3 cm-1 on the laser incident part of our sample at 750 oC, which are D, G, and 2D peaks, respectively. These are the characteristic peaks for carbon-based materials like graphene. We further raised the temperature of the resistive heater to 850 ℃ at an increment rate of 50 ℃ and found CBMs in both laser incident and non-incident areas. The Raman shifts for the D, G, and 2D peaks were all in the same ballpark. This means that at 20 sccm, we were able to commence the reaction by generating enough mass diffusion and flux from the gas phase and the substrate surface. Furthermore, the pulsed laser photons effectively raised the local temperature of the bombarded region. Under these deposition conditions, our CBMs were highly defective due to the predominance of the D peak's intensity compared to high-quality graphene.”
In the manuscript, table 4 shows the peak positions, intensity, and the number of layers obtained from Raman spectroscopy for each deposition. From the Table 4 below, we can conclude after taking ratios of the intensities that all the samples have many flakes/layers of graphene. The same information can be obtained if we take the ratio of the areas associated with the respective Raman peaks. Sample Cu15 is the commercial grade graphene and used as the gold standard for our deposition. Here we can see that it is a single layer graphene with zero defect. It can be seen from the D peak that sample Cu13,1 has the highest defect concentration, the defect data looks rather inconclusive, and more research needs to be done on this. Same is true for the carbon concentration shown from the G peak.
- In Raman spectra (Fig. 3(a-o), provide the data from 3000 cm-1. There is no appearance of D-band in provided data. How D and G band intensity varied with deposition temperature and CH4 gas flow?
Response: It was observed that our CBMs were highly defective because of the prominence of the intensity of the D peak when compared with high quality graphene in Figure 2. Sample Cu15 is the commercial grade graphene we purchased and used as the gold standard for our deposition. Here we can see that it is a single layer graphene with almost zero defect. It can be seen from the D peak that sample Cu13,1 (Fig. 4f) has the highest defect concentration, the defect data looks rather inconclusive, and more research needs to be done on this, same goes for the carbon concentration shown from the G peak.
9. How 2D band in Raman spectra related with thickness/ No of carbon layers? Provides discussion on this finding.
Response: We address the following concern in our manuscript in page-7#5-6; 8#1-7.
“Our main goal is to shorten the CBM deposition time and sharpen the 2D peak by reducing the number of layers. The deposition time was reduced from 5 to 3 and then 1 minute, but there was no significant improvement of the 2D peak. From this, it is concluded that it might not be graphene but rather a graphene-like material. It was observed that the deposited substrate was PyC. It is produced by heating a hydrocarbon (CH4), nearly to its thermal decomposition temperature of around 1000 -1200 ℃. This thermal decomposition temperature breaks its bonds, releasing carbon free radicals and allowing graphite to crystalize in the absence of oxygen (pyrolysis) [32]. The surface roughness of PyC films does not vary significantly with film thickness [33], and it is also responsible for quenching the 2D signal in samples Cu11,1, Cu12,1, and Cu13,1 regardless of the deposition time.”
10. Authors should provide the SEM images of deposited carbon materials and analyzed its surface morphology. Only Raman characterization in not sufficient in this study.
Response: We plan to perform the suggested experiments and analyses in future, and we will publish the data in the next phase of the research.
11. Provided the XRD pattern of carbon materials and check it crystallinity.
Response: We plan to perform the suggested experiments and analyses in future, and we will publish the data in the next phase of the research.
12. What was the amount of C and O in synthesized carbon materials? Check it by using EDS-SEM analysis.
Response: We plan to perform the suggested experiments and analyses in future, and we will publish the data in the next phase of the research.
13. There are some grammatical and punctuation errors in this manuscript. The English language should be improved. Tenses are not consistent from sentence to sentence and there are some grammatical errors.
Response: We have carefully revised the whole paper and the overall grammatical and punctuation errors are taken care of.
References:
[1] Johnson, C.E., Weimer, W.A. and Cerio, F.M., 1992. Efficiency of methane and acetylene in forming diamond by microwave plasma assisted chemical vapor deposition. Journal of materials research, 7(6), pp.1427-1431.
[2] Bhaviripudi, S., Jia, X., Dresselhaus, M.S. and Kong, J., 2010. Role of kinetic factors in chemical vapor deposition synthesis of uniform large area graphene using copper catalyst. Nano letters, 10(10), pp.4128-4133.
[3] Munoz, R. and Gómez‐Aleixandre, C., 2013. Review of CVD synthesis of graphene. Chemical Vapor Deposition, 19(10-11-12), pp.297-322.
Round 2
Reviewer 1 Report
The authors have satisfactorily addressed the raised concerns, therefore the article is recommended to be accepted for publishing.
Reviewer 2 Report
Accept